# S Protein, ACE2 and Host Cell Proteases in SARS-CoV-2 Cell Entry and Infectivity; Is Soluble ACE2 a Two Blade Sword? A Narrative Review

**DOI:** 10.3390/vaccines11020204

**Published:** 2023-01-17

**Authors:** Reza Nejat, Maziar Fayaz Torshizi, David J. Najafi

**Affiliations:** 1Department of Anesthesiology and Critical Care Medicine, Laleh Hospital, Tehran 1467684595, Iran; 2Department of Chemical Engineering, Imperial College London, London SW7 2AZ, UK; 3Alliance Retina Consultants, La Mesa, CA 91942, USA

**Keywords:** ACE2, Ang II, angiotensin(1–7), AT1R, cathepsin, COVID-19, endocytosis, ERK1/2, furin, IL-1β, IL-6, MAPK, MasR, NF-κB, PLC, PKC, RAS, RBD, SARS-CoV-2, S protein, TMPRSS2, TNF-α

## Abstract

Since the spread of the deadly virus SARS-CoV-2 in late 2019, researchers have restlessly sought to unravel how the virus enters the host cells. Some proteins on each side of the interaction between the virus and the host cells are involved as the major contributors to this process: (1) the nano-machine spike protein on behalf of the virus, (2) angiotensin converting enzyme II, the mono-carboxypeptidase and the key component of renin angiotensin system on behalf of the host cell, (3) some host proteases and proteins exploited by SARS-CoV-2. In this review, the complex process of SARS-CoV-2 entrance into the host cells with the contribution of the involved host proteins as well as the sequential conformational changes in the spike protein tending to increase the probability of complexification of the latter with angiotensin converting enzyme II, the receptor of the virus on the host cells, are discussed. Moreover, the release of the catalytic ectodomain of angiotensin converting enzyme II as its soluble form in the extracellular space and its positive or negative impact on the infectivity of the virus are considered.

## 1. Introduction

Deciphering the pattern of SARS-CoV-2 entry into the host cells, a crucial step in the virus infectivity, delineates a way to propose effective measures to interfere with the pathogenesis of COVID-19 successfully. Describing SARS-CoV-2 entry into cells by plainly describing the picking up of the virus, bound to its receptor, by the cells would be a rather simplification of a complicated process actually associated with the elaborate interactions among extra- and intra-cellular proteases, angiotensin converting enzyme II (ACE2) and SARS-CoV-2 spike (S) protein [1,2]. The docking of the S protein to its receptor, ACE2, a type I transmembrane mono-carboxypeptidase of renin angiotensin system (RAS), eventually leads to the fusion of the virus membrane with the host cell plasma membrane to release the viral genome into the cell cytoplasm [3]. Additionally, the fusion may occur at the membrane of endosomal compartment after clathrin-mediated endocytosis of the virus-ACE2 complex [4,5]. Moreover, there have been non-endosomal clathrin-independent models introduced for SARS-CoV-2 cell entry [4]. Although the fascinating ACE2-dependent entry has extensively been explored as the major model of SARS-CoV-2 entrance into the cells, other ACE2-independent models such as those mediated by Fc gamma receptors (FcγR) in monocytes have also been suggested [6,7,8,9]. It is also noticeable that the catalytic subunit of ACE2 released as a soluble molecule (sACE2) into the extracellular fluid with the ability to entrap the roaming virions in the circulation may negatively affect the infectivity of SARS-CoV-2 [10]. Paradoxically, sACE2 activity in the plasma was considered to have a positive correlation with the severity and mortality of COVID-19 [11]. In this context, the contribution of *A D*istegrin *a*nd *M*etalloproteinase 17 (ADAM17) to cleavage of ACE2 is worth investigating (see Section 4 and Section 9) [10].

Additionally, the priming effects of a constellation of proteolytic proteins in different models of SARS-CoV-2 cell entry should not be ignored [12], among those the transmembrane serine protease type II (TMPRSS2), cathepsins, furin and metalloproteases are of utmost importance [13,14,15,16]. To make it more complicated, the proteases themselves are secreted as inactive zymogens containing an inhibitor prodomain which covers the active site. The activation of proteases requires that this prodomain be dissociated in a meticulously regulated fashion [17]. The prodomain of TMPRSS2 is needed to be autocleaved [18] while in the case of cathepsins, cleavage of the inhibitory prodomain by distinct peptidases is pH-dependent [19]. In addition, furin requires pH- and Ca^2+^-dependent autoproteolytic removal of an 83-amino acid propeptide during its intracellular trafficking from the trans-Golgi network (TGN)/endosomal system to its destination on the cell membrane [20]. ADAM17, mostly residing in endoplasmic reticulum as an inactive metalloprotease, loses its prodomain through cleavage by a pro-protein convertase such as furin in Golgi apparatus [21].

In this review, the complex nature of SARS-CoV-2 cell entry with special attention to the role of proteinases and the updates of contribution of dysregulation of renin-angiotensin system (RAS) to the pathogenesis of COVID-19 at molecular and cellular level are discussed. Additionally, the paradox of the function of the soluble form of ACE2 and its relevant therapeutic implications are described.

## 2. S Protein Structure

S protein of SARS-CoV-2 (of 1273 amino acid residues), a class I fusion and homo-trimeric glycoprotein, is composed of two subunits, S1 and S2, in each of its monomers (Figure 1). The former is in charge of docking of the virion to ACE2 and the latter promotes fusion of viral and host cell lipid outer membranes through a fusogenic process [22]. From N- toward C-terminal ends, S1 comprises a signaling sequence, an N-terminal domain (NTD), a receptor binding domain (RBD) including its receptor binding motif (RBM) and two subdomains (SD1 and SD2) [23,24]. S2, a rather conserved molecule in SARS-CoV and SARS-CoV-2, consists of a fusion peptide (FP), the heptad repeat 1 (HR1), the heptad repeat 2 (HR2), a transmembrane domain (TM) and a cytoplasmic tail (CT) [22,25] (Figure 1). Absent in SARS-CoV, a polybasic amino acid sequence, RRAR (Arg-Arg-Ala-Arg), called the furin-cleavage site is located at the S1/S2 boundary, the deletion of which attenuates SARS-CoV-2 replication in the respiratory cells [26]. It has been shown that the presence of the RRAR motif enables SARS-CoV-2 to evade endosomal interferon-induced transmembrane (IFITM) proteins [27].

## 3. Binding of ACE2 with RBD Increases the Chance of Further Complexification of ACE2 and SARS-CoV-2

Binding of RBD with ACE2 is the first step of virus cell entry. Glycosylation of ACE2 and RBD residues may block this process [29]. However, RBDs in a trimeric S protein are not always accessible for this complexification and may be packed together [30]. Constitutively, the RBD position within the S1 structure shifts dynamically through an intrinsic hinge-like movement between “open or up (U)” and “closed or down (D)” states [23] (Figure 2). RBD in U status moves outward the apex of S1 domain to be ACE2-accessible while in D position it hides within the protein apex to be ACE2-inaccessible [31] (Figure 1 and Figure 2). As to the trimeric nature of S protein and considering U or D states of RBD, the apex of the trimeric S1 might harbor symmetric three open (UUU) or three closed (DDD) RBDs as well as intermediate asymmetric conformations of one open/two closed (UDD) or two open/one closed (UUD) ACE2-binding domains. The predominant RBD conformation (UUU, DDD or other intermediates) differs in the variants of the SARS-CoV-2 [32]. Recent studies attributed the escalated transmissibility and virulence of the new mutant variants to the abundancy and stability of RBD in the U state [32,33]. Moreover, according to cryo-electron microscopy findings, docking of the single open RBD to ACE2 induces a rotation of the same domain which provokes its center of mass within the UDD complex to move approximately 5.5 Å away from the axis of the whole trimeric S protein molecule. This rotation is accompanied by moving of the three NTDs in the trimer of about 1.5–3.0 Å [34] which results in a reduction in the contact area among S1 subunits and the neighboring S2 core. This causes successive gaining of U state in the neighboring two D-positioned RBDs within the trimeric S protein [34] (Figure 1). To make it more clear, it should be noticed that in ACE2-unbound form of closed RBD, there are π˗π interactions among some residues of S1 in one protomer and Tyr(Y)837 of S2 in the neighboring protomer in addition to a salt bridge between Asp(D)614 of S1 and Lys(K)854 of the S2 which are disordered after ACE2 binding [34]. In D614G (Asp614 to Glycine) mutation, the hydrogen bond between D614 residue of SD2 (of S1 subunit in one protomer) and Threonine(T)859 residue of S2 subunit in the neighboring protomer is deleted which induces S1 to move away from S-trimer axis [23,35]. These conformational changes bringing about successive expression of more U states results in escalating the probability of complexification, not the *affinity*, of RBD with ACE2 [35]. Clarifying this issue needs to consider that the binding affinity (according to its definition [36]) depends on the quantity and type of the mutations crowded in RBD which determine the density and dynamicity of polar interactions of RBM interfacial and non-interfacial amino acid residues with the counterpart residues of ACE2 [37,38].

## 4. Proteases Play a Pivotal Role in Various Models of SARS-CoV-2 Cell Entry

Various models of SARS-CoV-2 cell entry have been introduced [2,4,6] all of which end up with fusion of the virus membrane and the host cell plasma membrane [2]. Docking of RBD to ACE2 initiates an allosteric destabilization of the polybasic motif (RRAR) at S1/S2 interface. From this point on, proteolytic processing of S protein ultimately results in fusogenic configuration of S protein enabling virus membrane to intermingle with the host cell membrane [43,44,45]. An in vitro study revealed that serine proteases, furin and TMPRSS2, could enhance SARS-CoV-2 replication efficiency and cytopathology [46]. Furin (see Section 5) contributes to the processing of RRAR motif, the furin-cleavage site, at S1/S2 boundary of SARS-CoV-2 [47]. This cleavage has been correlated with syncytia formation of the infected lung epithelial cells [27]. Remaining non-covalently attached to S1 after the furin cleavage [48,49], S2 following formation of RBD-ACE2 complex is further proteolytically cleaved at S2′ segment located proximal to fusion peptide (FP) [34,50]. The cleavage at S2′ is mediated by the host cell plasma membrane serine protease (TMPRSS2, see Section 6) as well as other tissue proteases [51,52,53,54,55,56]. This promotes shedding of S1 subunit and exposure of FP [57]. Containing hydrophobic residues, FP is responsible for insertion of S protein into the host cell plasma membrane through gaining a wedge-shaped conformation in a Ca^2+^-dependent process [50,58]. Prior to fusion, sufficient energy is provided by a transition of the conformational shape of S protein from metastable prefusion into irreversible postfusion status [2]. This ultimately leads to a trimeric hairpin kneeling reconfiguration of HR1 and HR2 domains of S2 in order to bring the lipid membrane of the virus into the vicinity of the host cell plasma membrane [59,60,61].

Given that the presence of furin-cleavage site in S protein enhances TMPRSS2-mediated cell entry [27], different studies debate whether the cleavage at S1/S2 boundary is essential for SARS-CoV-2 penetration to the respiratory cells or cell-cell fusion as a means of direct spreading of the virus among neighboring cells [3,62]. A study on 293T cells infected with SARS-CoV-2, the presence of ACE2 but not TMPRSS2 was sufficient for syncytia formation albeit the destabilization of S1/S2 cleavage at the multibasic residues contributed to effective cell-cell fusion [63]. As a matter of fact, it was demonstrated that the presence of furin is not absolutely necessary for this proteolysis because other cellular proteases may also take over its function, yet furin enhances the process substantially [64,65]. Intriguingly, Tang et al., showed that S1/S2 cleavage is a prerequisite for TMPRSS cleavage of S2′ in case SARS-CoV-2 enters the host cell through the cell surface plasma membrane but not for the virus invasion via endosomal internalization [52]. For any route of the virus entry, the importance of the proteolytic destabilizing cleavage of S1/S2 junction (see Section 3, gaining U from D states in RBDs) and subsequent proteases-induced transition of S2 subunit to fusogenic configuration should be emphasized [66]. In silico and cell culture studies showed peptidomimetic binding compounds as well as lactoferricin and lactoferrin hinder the infectivity of SARS-CoV-2 significantly through inhibition of TMPRSS2-induced priming of S protein [67,68]. However, a recent study on HEK293T cells revealed that both TMPRSS2 and metalloproteases, particularly ADAM17, individually or co-operatively, promote activation of S2′ segment of S protein and syncytia formation in infection by SARS-CoV-2 and its emerging variants of concern [69]. Another study on various cell lines demonstrated cell-specific characteristic of this metalloprotease-dependent SARS-CoV-2 cell entry. This also requires the presence of furin-cleavage motif that is unique to SARS-CoV-2 not seen in SARS-CoV or MERS-CoV [70]. In addition, a cell free assay revealed that ADAM17 and ADAM10 (another member of ADAM family) are able to prime S protein, as well [71].

The endocytic cell entry of SARS-CoV-2 follows either a canonical (clathrin-mediated) or non-canonical (non-clathrin-mediated) pathway; the former proceeds through budding of clathrin-coated vesicles while the latter comprises caveolae-mediated, flotillin-1-dependent and glycosylphosphatidyl inositol-anchored protein endocytic pathways as well as macropinocytosis [4]. Moreover, the mandatory well-orchestrated role that the host Cathepsin B/L play in activation of S protein in both endosomal clathrin-dependent or non-endosomal clathrin-independent routes of entry must not be ignored, either [5,6,14,72]. In these two endocytic processes, activation of Cathepsins needs endosomal acidification interlinked with NAADP-elicited Ca^2+^ loss from the endosome which is mediated by vacuolar H^+^-ATPase (V-ATPase) [73,74,75,76,77].

## 5. Furin, a Ubiquitously Expressed Multifunctional Protease

Furin, (a type I transmembrane serine endopeptidase) trafficking between plasma membrane and endosomes is reserved predominantly in trans-Golgi network (TGN) and contributes to a quite large array of human physiologic, pathologic and metabolic processes as well as bacterial (anthrax) and viral (avian influenza, HIV) pathogenesis [78,79,80,81]. This endopeptidase comprises a signal peptide, an inhibitory prodomain, a subtilisin-like catalytic domain, a middle P domain, a cysteine-rich domain, a transmembrane peptide and a cytoplasmic tail. Full activation of furin, as a pre-proprotein, requires that the molecule be autocleaved at a primary (Arg-Thr-Lys-Arg^107^↓Asp^108^) as well as a secondary (Arg-Gly-Val-Thr-Lys-Arg^75^↓Ser^76^) site; the latter results in the release of the inhibitory prodomain fragment [82].

Furin exists in two forms: a membrane-bound and a circulating soluble form secreted by the cells; the higher levels of the latter increases the risk of metabolic syndrome, diabetes and hypertension [83]. This enzyme, ubiquitously expressed in tissues, is a member of Ca^2+^-dependent prohormone/proprotein convertase (PC) family involved in the proteolytic maturation of many hormones, neuropeptides, immune system molecular effectors, growth factors, von Willebrand factor, receptors (e.g., Notch receptors) and other enzymes such as ADAM17 (see Section 9) and metalloproteinases (MMPs) [84,85,86,87,88]. Furin proteolytic modification of its substrates occurs through cleavage at the C-terminal end of multibasic motifs (Arg-*X*-Arg/Lys/Arg-Arg↓ homologous to RRAR residues at S1/S2 boundary; see Section 2) in the substrates in the presence of Ca^2+^ [89]. Through this cleavage of the similar motif at S1/S2 interface of SARS-CoV-2, furin plays a pivotal role in the virus infectivity/pathogenicity [45]. Revealing the importance of this process, a cell culture study demonstrated that host cell-mediated *O*-glycosylation of S protein at proline681 (P^681^) proximal to the RRAR motif weakens furin-mediated cleavage and attenuates SARS-CoV-2 infectivity. Mutation of P^681^ in alfa and delta variants of this virus which mitigates *O*-glycosylation increases furin cleavage and potentiates the distribution of the virus and syncytia formation [90]. Moreover, a mutant variant of SARS-CoV-2 with deleted furin cleavage site showed attenuated infectivity, yet this variant compared to the wild type virus needed higher levels of anti-RBD antibody to be neutralized. This negates its usefulness for manufacturing an effective vaccine [91]. In a clinical study on a limited number of patients with COVID-19, the level of circulating furin was found to be correlated with the disease severity [92].

Optimum pH for full activation of furin is 7.0 though more than 50% of its maximum activity, depending on the substrates, occurs at pH between 6.5 and 8.0 [93]. Moreover, furin requires a calcium concentration of 10^−3^ M for its full function [79]. Assuming that extracellular ionized Ca concentration is about 10^−3^ M compared to its cytoplasmic content of about 10^−7^ M (reaches to 1–5 × 10^−4^ M in intracellular Ca^2+^ stores or escalates ultimately to 10^−6^ M to 10^−5^ M in stimulation of cell signaling [94,95]), it is obvious that nearly full proteolytic activity of furin in exocytic and endocytic pathways occurs extracellularly or in TNG/endosomal compartments and not in the cytoplasm [79,93,96]. In addition, a soluble active form of furin has long been recognized [97]. Given that the favorable ionized Ca2+ concentration for furin function is provided in extracellular space, it is implied that soluble furin isoform might potentiate SARS-CoV-2 S protein cleavage in a juxtacrine fashion [98].

## 6. TMPRSS2, Its Soluble Protease Domain and Calcium

TMPRSS2, a transcript product of an androgen-regulated polymorphic gene, *tmprss2*, [99,100] is expressed on the plasma membrane of several cell types [101]. This serine protease exists as two isoforms (1 and 2); the former contains 529 (with an extended cytoplasmic N-terminus) and the latter 492 residues [102]. TMPRSS2 co-expressed with ACE2 in the lungs and some other tissues, plays a pivotal role for entrance of SARS-CoV and SARS-CoV-2 into the host cells through priming the virus S protein (see Section 2 and Section 5) and cleavage of ACE2 (see Section 8) [53,103]. TMPRSS2 as a type II transmembrane protease is composed of an intracellular N-terminus, a single pass transmembrane domain and an extracellular segment. The latter contains three domains: a low density lipoprotein receptor type A (LDLRA), a class A scavenger receptor cysteine-rich (SRCR) and the C-terminal canonical serine protease domain (PD_TMP_) [102,104] (Figure 3).

Each of LDLRA and SRCR domains, collectively called the stem region which contributes to substrate recognition as well as protein-protein interactions and ligand binding, harbor Ca^2+^ binding sites [102,105,106]. PD_TMP_ cleaves a variety of peptide substrates at Arginine and Lysine residues [106] which in the case of ACE2 occurs at the cluster of these two amino acids located in residues 697–716, noticing that this proteolysis is required for TMPRSS2-mediated cathepsin-independent virus entrance to the host cell [107]. Moreover, activation of TMPRSS2 (isoform 2) requires autocleavage of its own molecule at Arg255 in an as yet obscure condition, which may result in the release of PD_TMP_ as a soluble form (sPD_TMP_) into extracellular space [108,109]. However, the majority of PD_TMP_ remains linked to the rest of the molecule through a disulfide bond [102,110]. sPD_TMP_ may interact with other cell surface proteins (in an autocrine manner), the extracellular matrix and proteins located on the neighboring cells (juxtacrine manner) [102]. Mutagenetic studies have demonstrated that SRCR, but not LDLRA, plays an essential role in proteolytic activity of PD_TMP_. Therefore, it might be a debatable issue whether PD_TMP_ after dissociation from the stem region retains its proteolytic function or gains a yet unraveled distinct activity [111]. It is worth reminding that Ca^2+^ binding to or unbinding from a distinct domain of a protein determines the task the domain should fulfil [112] or affects the structural stabilization of the proteinases and substrates affinity [113,114]. It means that this bivalent ion participates in the regulation of the functions of SRCR and LDLRA domains rather than the proteolytic activity of PD_TMP_ by itself after the latter is released as sPD_TMP_ to the extracellular space losing its support by SRCR.

## 7. Major Activity of Cathepsins Occurs in the Membrane-Bound Intracellular Organelles

Human cathepsins, categorized in three families and 15 classes, function as lysosomal proteases and contribute to many physiological processes. These proteases are involved in the process of autophagy, the disequilibrium of which leads to numerous pathologies. Cathepsin activity is pH-dependent: acidic pH (3.5–6) enhances while neutral pH attenuates and alkaline pH inactivates their function [115]. However, some cathepsins including cathepsin K and H have been found to function at pH 7 in a stable manner implicating their ability to display activity outside lysosomes [116]. However, the majority of cathepsins, including cathepsin B and L that contribute to the activation of the S protein, undergo processing and maturation in the lysosomes rather than within the cytoplasm or extracellular space, otherwise they might induce apoptotic or necroptotic pathways or provoke inflammation in addition to their physiological role in tissue repair, extracellular matrix degradation and remodeling [115,117]. Even though TMPRSS2 and cathepsins may both contribute to activation of SARS-CoV-2 S protein, the former displays a more dominant function than the latter [118].

## 8. Membrane-Bound and Soluble ACE2

ACE2 contributes to infectivity and the entry of SARS-CoV-2 into the host cells while physiologically plays a crucial role in protection of the lungs, heart and other tissues against a variety of inflammatory or hypoxia-induced insults [119,120,121,122]. The non-homogenous distribution of ACE2, yet in a limited scale compared to ACE, was shown in a wide variety of tissues with the highest expression in the small intestine, heart, testis, thyroid glands and adipose tissue; intermediate expression in the lungs, colon, liver, bladder and adrenal glands; and the lowest content in the blood, spleen, bone marrow, brain, blood vessels and muscles [123]. Beyond the genetic traits determined by loci near immune related genes such as *IFNAR2* and *CXCR6* [124], the distinct genetic variants of highly polymorphic *Ace2* (gene located on chromosome Xp22.2 encoding for ACE2 protein [125]) in different populations have also been shown to affect the susceptibility to and severity of COVID-19 in some cohorts of patients [126,127,128,129].

ACE2, a type I transmembrane protein of 805 residues, comprises an extracellular segment (residues 19–740) including a signal peptide of 18 amino acids, an N-terminal claw-like protease domain (PD_ACE2_, residues 19–615) and a collectrin-like domain (residues 616–740) including ferredoxin-like fold “Neck” domain (residues 616–726) attached to a long transmembrane domain (residues 741–763) ending with a cytosolic C-terminal tail [130,131,132,133,134] (Figure 4). The cytosolic tail containing a conserved endocytosis short linear motif had been demonstrated to play a role in SARS-CoV-S protein-induced shedding of ACE2 (see Section 9), TNF-α production and SARS-CoV infection [135,136] although a combined in silico and in vitro study strengthened by confocal imaging denied any role for C-terminal tail of ACE2 in SARS-CoV and SARS-CoV-2 cell entry [137]. The collectrin-like domain contributes to dimerization of two ACE2 molecules (assuming ACE2-A and ACE2-B) through interacting with Arg652, Glu653, Ser709, Arg710 and Asp713 in ACE2-A with Tyr641, Tyr633, Asn638, Glu639, Asn636 and Arg716 in ACE2-B [130].

Physiologically, PD_ACE2_ with its catalytic site removes one amino acid from the C-terminal end of angiotensin I (Ang I) and angiotensin II (Ang II) to turn them into angiotensin (1–9) and angiotensin (1–7), respectively [138]. An X-ray study uncovered the presence of a wide and deep cleft within the metallopeptidase catalytic domain of ACE2 with two subdomains I and II comprising its wall [139]. The catalytic efficiency of ACE2 for hydrolysis of Ang II is 400 times more than that of Ang I (as the precursor of Ang II) [138]. This means that any deficiency of ACE2 affects hydrolysis of Ang II 400 times more than that of its precursor, Ang I. Physiologically, this enzymatic hydrolysis leads to an increase in angiotensin(1–7)/Ang II concentration ratio resulting in attenuation of proinflammatory and organ-damaging effects of Ang II/AT1R pathway [140]. Conversely, deficiency of ACE2, which leads to an increase in tissue or plasma Ang II, may deprive the cell of one of its anti-inflammatory and protective tools [119]. An animal study on *Ace2* (ACE2 gene) knockout mice proved that cardiac contractility is impaired, hypoxia-inducible genes such as *BINP3* (encoding pro-apoptotic Bcl2 interacting protein) are upregulated and Ang II *increases* in the kidneys, heart and plasma [141,142]. Reciprocally, Ang II, while upregulates angiotensin converting enzyme (ACE), was shown, both in vitro and in vivo, to downregulate ACE2 expression at both mRNA and protein levels rather through a cellular inflammatory reaction induced by AT1R mediated ERK/p38MAPK pathway [143,144]. Furthermore, as a positive feedback effect, Ang II through stimulation of AT1R promotes shedding of the membrane bound ACE2 (mACE2) ectodomain off the cell surface into the plasma as soluble ACE2 (sACE2; residues 18-708) [145] (see above in this section) by inducing a sheddase protein, ADAM17 (see Section 9) [146,147]. Both of these two effects, which results in a decrease in mACE2 and an increase in circulating sACE2, can be blocked by Ang II type I blockers (ARBs), e.g., losartan [143,144,147]. Consequently, the concentration of sACE2 increases in pathologies associated with higher AT1R stimulation such as myocardial infarction, heart failure, metabolic syndrome and diabetes mellitus [148,149,150].

## 9. ADAM17, the Sheddase of ACE2

Regulated selective cleavage and release of the extracellular segment of many of transmembrane proteins into the extracellular space called “ectodomain shedding” modifies a diverse array of trans- and cis-signaling pathways [151]. ADAM17, the first identified shedding protease, is a Zn^2+^-dependent metalloproteinase and a member of the family of membrane-anchored ADAMs, which plays a role in both innate and humoral adaptive immunity, among the others [152]. This sheddase promotes proinflammatory effects through dissociating a variety of ligand proteins including ACE2, TNF-α and transmembrane CX3CL as its substrate as well as transmembrane receptors including IL6Ra and TNF receptors I and II [146,153,154,155]. This metalloprotease exists in two forms: the full length pro-ADAM17 of 100KDa and the mature form of 80KDa lacking the inhibitory prodomain: the latter comprises almost two thirds of its total cell content which resides predominantly in perinuclear space along with TNF-α [156,157]. ADAM17 as a type I transmembrane multidomain proteinase comprises an N-terminal signal sequence, an inhibitory prodomain, a metalloproteinase catalytic domain, a disintegrin-like domain, a cysteine-rich and membrane proximal domains (MPD, involved in substrate recognition) attached to a single transmembrane domain ending into a cytoplasmic tail [158,159] (Figure 5). The transmembrane domain, but not the cytoplasmic domain, contributes to the rapid activation of this metalloprotease by different signaling pathways [160]. However, phosphorylation of the cytoplasmic domain of ADAM17 by mitogen activating protein kinase (MAPK) network including p38MAPK and extracellular signal-regulated kinase (ERK) as well as Polo-like kinase2 (PLK2) keep this metalloprotease proteolytically functional through prevention of its dimerization and dampening of its binding to tissue inhibitor of metalloproteinase3 (TIMP3) at the cell surface [161,162]; TIMP3 plays an inhibitory role for the dimerized ADAM17 on the cell surface [163].

A proteolytically inactive rhomboid protein (iRhom2) encoded by *Rhbdf2* gene involved in innate immunity against viral infections [165], regulates trafficking of pro-ADAM17 from endoplasmic reticulum to Golgi apparatus where furin detaches ADAM17 inhibitory prodomain to make it mature, substrate specific and be prepared for expression on the cell surface in complex with iRhom2 [166,167]. Infection with DNA and RNA viruses (influenza A, RSV) upregulates *Rhbdf2* and *ADAM17* genes [167]. However, an in vivo study revealed that ADAM17-dependent ectodomain shedding is a saturable phenomenon: it does not increase significantly when ADAM17 expression rises above a certain level [168]. It must be mentioned that this metalloprotease is rapidly and reversibly switched “on” and “off” through conformational changes making its catalytic domain accessible and inaccessible, respectively, yet independent of removal of the inhibitory pro-domain or dissociation of TIMP3 [169]. 

Given that ADAM17 small interfering RNA (siRNA) is capable of attenuating Ang II-mediated inflammation in vascular smooth muscle cells [170] it is implied that along with other downstream mediators, ADAM17 is also involved in Ang II-induced inflammatory responses. Additionally, Ang II-mediated stimulation of AT1R, a G-protein coupled receptor (GPCR), provokes oxidative stress, induces phospholipase C (PLC)-mediated protein kinase C (PKC) activation and increases calcium influx [171,172,173,174]. Oxidative stress in tumor cells and platelets could previously be found to activate ADAM17 with pro-inflammatory effects (see previous paragraph) [175,176]. Reciprocally, ADAM17 in a mouse model could also increase NADPH oxidase 4 (Nox4) activity resulting in oxidative stress [177] which by inducing pro-inflammatory genes is intertwined with inflammation [178,179]. It is, however, noticeable that ADAM17 positively regulates Thioredoxin-1 (Trx-1) activity as the key effector of intracellular reducing system and downregulates ADAM17 as a negative feedback effect [180,181].

Furthermore, the PKC pathway, depending on the nature of its activator, affects ADAM17-dependent ectodomain shedding. Short-term (minutes) activation of protein kinase C (PKC) by phorbol ester (the strongest non-physiologic stimulator of PKC) increases ADAM17 content on the cell plasma membrane and the sheddase activity while prolonged (hours) exposure to phorbol ester downregulates mature form of ADAM17 on the cell surface abolishing the ectodomain shedding without reducing the total cell content of ADAM17 (mature plus pro-ADAM17) [21]. Conversely, ligand-activated GPCRs, such as thrombin-mediated protease activated receptor1 (PAR1), with the potential to activating of PKC signaling [182,183], induces sheddase function without any change in the content of ADAM17 on the plasma membrane [21]. Moreover, Ang II, via AT1R, was found to upregulate PAR1 and PAR2 in rat aorta which lead to pro-inflammatory responses accompanied by raising IL-6 and monocyte chemoattractant protein-1 (MCP-1), the substrates of ADAM17 [156,184,185]. Along with the natural homo- and hetero-merization of GPCRs [172], an in vitro study revealed synergistic interaction between AT1R and PAR1 [186], implicating a positive influence of these receptors in activating ADAM17. Moreover, AT1R stimulation alone is associated with a rise in IL-1β which by itself promotes ectodomain shedding by ADAM17 [187,188]. Considering that Ang II via AT1R could induce PKCδ/p38MAPK [189] which also activates ADAM17 [190,191], it is logically implied that AT1R also potentiates ADAM17-induced ectodomain shedding through p38MAPK pathway.

Ca^2+^ influx and calmodulin inhibition stimulate ADAM10 [192]. Given that ADAM17 is involved in processing of pro-α2δ-1 and α2δ-3 subunits of voltage gated calcium channels which enhances calcium influx [193] it is implied that ADAM17 also indirectly contributes to activating ADAM10 in ectodomain shedding.

## 10. Shedding of ACE2 Ectodomain and sACE2 in SARS-CoV-2

As was mentioned previously, ADAM10 and ADAM17, the predominant sheddases, contribute to cleaving mACE2 off the cell surface which lead to a vast array of physiological and pathological cis- and trans-signaling [155]. Cleavage of the ectodomain of mACE2 occurs through a basal low level constitutive or a metalloproteinase-dependent fashion [194]. A cell culture study showed that inhibition of both ADAM17 and ADAM10, widely found on pneumocyte type I and II, results in ACE2 increment on the surface of these cells [71]. Consistently, an animal model of diabetes proved that ADAM17 gene knockdown and overexpression could reduce and raise mACE2 shedding off the cardiomyocytes, respectively [195].

Oxidative stress, MAPK network, PKC and Ca^2+^ signaling activation in various viral infections including SARS-CoV and SARS-CoV-2 has already been described [196,197,198,199,200,201,202,203]. Ang II-mediated AT1R stimulation activates these signaling pathways, as well [204,205]. All these signaling pathways directly promote ADAM17-dependent ACE2 shedding which by itself decreases angiotensin(1–7)/Ang II concentration ratio and promotes Ang II/AT1R pathway which also induces ADAM17 sheddase function (see previous section). Additionally, COVID-19 has been shown to promote ADAM17 expression both at the protein and transcriptional level [206]. 

In this context, neither should it be forgotten that the downregulation of mACE2 in COVID19 can be due to internalization of the virus-ACE2 complex [207,208,209,210] nor should the cleaving effects of TMPRSS2 along with ADAM17 be ignored. These two proteases compete in shedding, yet at different sites, of ACE2: ADAM17 and TMPRSS2 cleave Arginine and Lysine amino acids in residues 652–659 and 697–716, respectively [107]. However, TMPRSS2-mediated dissociation of mACE2 does not result in the release of sACE2 [10].

It was also shown that docking of the SARS-CoV S protein with ACE2 could induce detachment of the latter off the cell membrane by ADAM17, which can be blocked by the TNF protease inhibitor 2 (TAPI-2) [135]. Concordantly, SARS-CoV infection was shown to downregulate mACE2 in Vero E6 and the cells of the lungs in mice [211,212]. Anti-ADAM17 compounds after SARS-CoV infection could, in vitro and in vivo, inhibit ACE2 shedding, reduce sACE2 and suppress infectivity of the virus [213]. Consistently, a cell culture study confirmed that SARS-CoV-2 S protein induces metalloprotease-dependent ACE2 shedding [69]. Kornilov et al., in the early months after the COVID-19 pandemic, considered the elevated plasma level of sACE2 as a *biomarker* or *possible* cause of more severe SARS-CoV-2 infection seen in pathologies associated with higher levels of sACE2, e.g., diabetes, heart failure, metabolic syndrome and myocardial infarction as well as in the old, men and postmenopausal women [214]. Whatever the mechanism is, cleavage of mACE2 reduces its presence on the cell membrane and raises its soluble form (sACE2, containing PD_ACE2_ including RBD docking domain) in extracellular space including airway surface liquid covering the lung epithelial cells [215,216].

## 11. sACE2: Attenuating Inflammatory Responses?

Both classical (ACE/Ang II/AT1R) and its counter-acting non-classical (ACE2/angiotensin(1–7)/Mas receptor) pathways of RAS exert their effects as an autocrine or paracrine signaling system at the local tissue level and also as an endocrine system signaling in the whole body [217]. In sepsis, upregulation of the classical endocrine RAS pathway to a definite level leads to life-saving responses, yet overstimulation of the local arm of the same classical RAS aggravates tissue inflammation which might result in acute respiratory distress syndrome (ARDS) attenuated through counter-activation of non-classic arm of local RAS [218]. Similarly, growing evidence suggests the contribution of dysregulated RAS to the pathogenesis of COVID-19 [219]. Plasma Ang II has been reported to increase early in critical SARS-CoV-2-induced lung injury which reaches high levels on admission and decreases steadily during the later days till discharge of the patient. It shows a positive correlation with the severity of the disease [220,221,222,223]. Furthermore, as sACE2 retains its catalytic activity [215] its coincidental increment with the decreasing of circulating Ang II level might be regarded as a sensible physiological response attenuating the adverse pathological effects of early overwhelming surge of Ang II. Concordantly, Lier et al., in a limited number of critical COVID-19 patients discovered that plasma concentration of sACE2 and angiotensin (1–7) increases while circulating Ang II decreases. They explained these to be likely the result of raised detachment of pulmonary cell mACE2 and its release into the circulation as sACE2 [224]. Similarly, in a recent study, sACE2 and Ang II were found to increase and decrease, respectively in severe COVID-19 [225].

Additionally, Kragstrup et al., in a study on COVID-19 patients determined the association of increasing sACE2 plasma content in the first week with the severity of the disease which could predict the worse outcome in 28 days post-infection. They concluded with uncertainty that increased sACE2 could play a role in spreading the virus to uninfected tissues [226]. Almost consistently, the findings of a recent human study of COVID-19 demonstrated that shedding of mACE2 was higher in all the patients compared to healthy controls on the day of admission and soared significantly to much higher levels in severe than moderate groups on the first post-admission week [227]. In this context, raised sACE2 may also be the result of ADAM-17 overactivation in more severe cases of COVID-19 which leads to an exceeding amount of mACE2 shedding. In a transgenic mouse model, monoclonal antibody against ADAM17 potently raised SARS-CoV-2 viral load due to decreased shedding of mACE2 which along with retaining anti-viral defense capabilities through increased *Ifnb* and *Isg56* genes expression, remarkably ameliorated the post-infection inflammatory responses [228]. This is implied that COVID-19-induced lung-destructive inflammation might erupt rather via aggravation of ADAM-17-mediated release of mACE2 and TNF-α which results in the dysregulation of local RAS in favor of Ang II/AT1R axis and promoting inflammatory responses.

It is noticeable that Patel et al., found persistently elevated sACE2 *activity* in recovered patients from 35 to 114 post-COVID-infection days which was positively correlated with the severity of the disease [229]. Nevertheless, Lundström et al., reported that sACE2 *concentration* increases in COVID-19 patients compared to healthy people (5.0 vs. 1.4 ng/mL; 3.5 fold). They also found that early increment in plasma sACE2 content in the first 14 days of COVID-19 was transient and decreased significantly within four months post-infection, yet remained more than that of the control group. They also described a positive correlation between raised sACE2 and the prolongation of the symptom, yet with insignificant association with the mortality, severity indices (care level, any need for ventilator support) or total number of comorbidities [230]. Intriguingly, the rise in sACE2 concentration in this study showed no meaningful relationship with the plasma content of CRP, IL-6, TNF-α and plasminogen activator inhibitor-1 but weakly to moderately correlated with monocyte count, vWF concentration, D-Dimer and coagulation factor VIII [230]. It is evident from Lundström’s and another studies that, in COVID-19, sACE2 concentration rises (3.5 fold) less than that of IL-6 (about 6 fold up to a mean of 32 pg/mL [230] compared to 5 pg/mL in healthy individuals [231]). Considering over-activation of ADAM17 in COVID-19 [10] and its role in both shedding of mACE2 and IL-6-induced inflammatory responses [232] it seems that the lack of strong correlation between sACE2 level and inflammatory markers (IL-6) might be due to the saturable nature of ADAM17-dependent shedding of mACE2 (see Section 9) [168]; sACE2 does not rise to the level of increasing the activation of ADAM17 which is responsible for the raised IL-6-dependent signaling, as well. Consistently, in a mathematical model it was demonstrated that high levels of ADAM17 activity could only modestly deplete ACE2 off the surface of pancreatic β-cells [233]. Moreover, COVID-19-induced inflammatory responses erupts rather through multiple cascades ending into a multifaceted vasculopathic cytokine upregulation [234,235].

Interestingly, Daniell et al., discovered that despite the more abundant sACE2 in the plasma, its *activity* (in both plasma and saliva) and plasma angiotensin(1–7) *concentration* decreased significantly in COVID-19; the activity recovered gradually through the time [236]. The authors found a moderate correlation between the decline in angiotensin(1–7) concentration and lower sACE2 activity and attributed the discrepancy between plasma sACE2 *content* and *activity* to the presence of some other unknown mechanisms which may negatively regulate this soluble carboxypeptidase *activity*, mostly probable due to binding of sACE2 with the circulating SARS-CoV-2 S protein [236]. However, as fluorogenic Mca-APK(Dnp) mimicking Ang II is not a selective substrate for ACE2, using this compound in this study might have diminished the accuracy of the activity assay [237]. Moreover, Mca-APK(Dnp) based ACE2 function assay may generate conflicting results due to the presence of NaCl in the media preparation [238]. In addition, a reduction in plasma angiotensin(1–7) may also result from its hydrolysis via plasma angiotensin converting enzyme (ACE) or other peptidases rather than due to provisional diminished sACE2-dependent Ang II hydrolysis [239,240]. Moreover, there are some concerns about the specificity of angiotensin (1–7) assays using commercially available ELISA tests and the method of preparation of the plasma sample [241]. It is also noticeable that sACE2 catalytic activity is not expected to vanish after docking with RBD as the active enzymatic and RBD-binding sites in this soluble carboxypeptidase are located far from each other [216]. However, this concept has also been debated because docking to RBD could slightly close the functional cleft (see above in ACE2 section) within the ACE2 catalytic domain to some degree, hence this might modestly reduce its peptidase activity [242] (Figure 4 and Figure 6). Nevertheless, Lambert et al., demonstrated the catalytic ability of sACE2 [194] and beyond this Kiselev et al., revealed that docking to SARS-CoV-2-RBD potentiates multifunctional ACE2 ability to catalyze des-Arg^9^-bradykinin, with more modest accelerating effect on hydrolyzing Ang II [243]. It should also be mentioned that some inaccuracies might have inadvertently occurred as Mca-APK(Dnp) used in Kieslev’s study is a non-specific substrate of ACE2 [244]. However, it shows that sACE2 in docking with RBD, also attenuates inflammatory responses not only through hydrolyzing Ang II but via catalysis of des-Arg^9^-bradykinin which is involved in inflammation through kinin B1-receptor [245,246].

Altogether, it is implied that the rise in sACE2 in COVID-19 is the result, not the cause, of SARS-CoV-2-induced activation of a cascade of proinflammatory pathways accompanied by ADAM-17 stimulation which lead to the release of mACE2 as a soluble form into the circulation. It seems to act as a protective means to attenuate systemic classical ACE/Ang II/AT1R proinflammatory pathway in COVID-19.

## 12. sACE2: Increasing SARS-CoV-2 Infectivity?

Cytoplasmic tail signaling of mACE2 was found to have no influence on SARS-CoV-2 cell entry [137]. Retaining the amino acid residues involved in docking to RBD, sACE2 lacking membrane anchors is able to bind with the S protein of SARS-CoV-2 floating in the extracellular space [216]. This seems to prevent circulating SARS-CoV-2 from attaching to mACE2 on the surface of the encountered cells [249,250]. On the other hand, it has been hypothesized that sACE2 contributes to the entry of circulating SARS-CoV-2 into the cells with lower mACE2 content [251]. Although may be fascinating, both negating and supporting reports have made this issue unresolved [216,252].

Yeung et al., discovered that sACE2-S protein complex may facilitate SARS-CoV-2 cell entry through receptor-mediated endocytosis which is much more enhanced by the fusion of this complex with AT1R, vasopressin or vasopressin receptor, AVPR1B. They also demonstrated that sACE2 in low concentration (up to 100 ng/mL) dose dependently increases and in high concentration (25–100 μg/mL) decreases SARS-CoV-2 cell entry, in vitro [253]. Their findings support several reports showing the efficacy of recombinant human sACE2 or ACE2-containing exosomes in inhibiting SARS-CoV-2 cell entry via blocking RBD to bind with mACE2 [254,255].

Attaching at the “Neck” domain, mACE2 molecules naturally form homodimers with higher binding affinity (12–22 nM) to RBD compared to its monomer (77 nM) [256,257] (Figure 7). Assuming the trimeric nature of the S protein and homodimeric configuration of mACE2, it is mathematically speculated that interaction of two trimeric S proteins and three dimeric mACE2 should be involved in full fusion of SARS-CoV-2 and its receptor [258]. As ADAM17 cleaves mACE2 particularly at residues 652–659 [107] included in the Neck-domain (residues 616–726) [247], it is not unexpected, yet needed to be studied, that sACE2 exist in a monomeric form. It has been discovered that as sACE2 monomer dissociate rapidly from RBD, occupation of the second and the third RBDs in the trimeric S protein by homo-dimer and homo-trimer of this soluble carboxypeptidase neutralizes the virus more effectively [249,257]. This encouraged experts to produce a stable mutant dimeric and a homo-trimeric variant of sACE2 with preserved peptidase activity but higher binding affinity to RBD (about 600 pM for the dimeric and 60 pM for the trimeric sACE2) to hinder mACE2-RBD complexification [257,259]. In order to block S proteins on each SARS-CoV-2 particle with high efficiency, multimerized sACE2-fusion proteins have been manufactured through molecular engineering [260].

A combined hydrogen/deuterium exchange mass spectrometry (HDX-MS) and molecular dynamic simulation study revealed flexibility and dynamics of some distal hotspots in S protein relative to the site of RBD-ACE2 complex [261]. Concordantly, mACE2 complexed with RBD in “up” conformation was found to change the dynamics of the RBD leading to a continuous swing motion of S1-mACE2 complex on the top of the trimeric S protein. This declines the surface between S1 of mACE2-attached protomer and S2 subunits of the other two unattached protomers leading the trimeric S protein to untwist (Figure 1 and Figure 2) so that the two closed RBDs sequentially gain “up” status [34,262] (see also Section 3 and Figure 2). This continuous swing motion also enhances RBD and NTD dynamics in the attached S1 and leads to transition of S2 subunit of the same protomer toward the stable postfusion status to facilitate SARS-CoV2 cell entry [262]. It is needed to study whether mACE2 point of attachment to the plasma membrane plays a role as a fulcrum in this scenario and whether docking of monomeric sACE2 to RBD might induce the same swing motion and conformational changes.

## 13. Clinical Evidence Showing Benefit of Recombinant Human Soluble ACE2

Reminding that SARS-CoV-2 cell entry eventually leads to downregulation of mACE2, it is supposed that administration of sACE2 results in suppression of mACE2 shedding through blocking RBD docking sites [263]. Hence, it is persuasively expected that recombinant human soluble ACE2 (rhsACE2) is able to rebalance local Ang II/angiotensin(1–7) ratio in favor of non-classical anti-inflammatory ACE2/angiotensin(1–7)/MasR axis to reduce massive release of destructive cytokines [264,265,266]. During the last decade, pharmacodynamic and pharmacokinetic characteristics of rhsACE2, its safety in healthy people [267] and subsiding effect in acute respiratory distress syndrome (ARDS) have been extensively studied [268,269,270].

There are several reports including the ones published by Roshanravan et al. [271] and Basit et al. [272], which show the promising effect of sACE2 (as a decoy receptor) or truncated ACE2 (amino acid sequence 21–119 of N-terminal region) on reducing virulence of SARS-CoV-2 and its variants of concern in preclinical experiments [273,274,275]. Even some experts tried to produce sACE2-IgG conjugate to overcome the short half-life of sACE2 in the circulation and to improve its activity against SARS-CoV-2 infection [276]. In another study, Zhang et al., found that rhsACE2-Fc created by fusion of rhsACE2 to crystallizable fragment (Fc) of the N-terminal of human IgG with long half-life and high affinity to SARS-CoV-2-RBD could dose-dependently neutralize SARS-CoV-2 and its variants of concern both prophylactically and therapeutically. No infection promotion was observed after administration of this product [277]. Furthermore, rhsACE2-Fc could hinder S-protein-dependent giant cell formation and protect human bronchial cells against the invasion of the virus. This product could potently suppress the invasion of the mutants of SARS-CoV-2 generated through sequential passage in mammalian cells [277]. A recent animal study revealed how the impact of RAS dysregulation due to loss of mACE2 which leads to oxidative stress might eventuate into lung injury through activation of Ang II/AT1R/NF-κB/NOX1&2 signaling pathway [278]; the deadly signaling cascade that had hypothetically been proposed as the pathophysiologic basis of lung inflammation in early weeks of SARS-CoV-2 pandemic [279]. According to this study, rhsACE2 was found to be able to cleave Ang I and Ang II to angiotensin (1–9) and angiotensin (1–7), respectively. Moreover, this product could diminish proinflammatory factors such as TNF-α, IL-1β, IL-6 and myeloperoxidase activity in broncho-alveolar lavage fluid and ameliorate SARS-CoV-2-RBD-aggravated acute lung injury [278]. Additionally, considering the mouth as the most accessible gate of entry of SARS-CoV-2-contaminated droplets, an ACE2-containing chewing gum has successfully been tried to reduce the high viral load in the tongue, buccal and gingival epithelial cells through entrapping the virions in the saliva and hindering the attachment of RBD with mACE2 on the mouth epithelial cells [280].

## 14. Future Directions

It is evident that precise recognition of fascinating SARS-CoV-2 entry into the cells, as the key component of its infectivity, provides a vast array of data enriching scientists to defeat against the incessant threat of this deadly virus. The vaccines, even though showed appreciable efficacy to reduce the worldwide burden of the COVID-19 pandemics, may lose anti-SARS-CoV-2 neutralizing potency in long term as the rate of the virus mutation in S protein hotspots against which vaccines are targeted, overwhelms the laboratories’ potentials to release up to date immunogenic, yet safe, particles. In addition, the ability of health care organizations to disseminate the vaccines evenly around the world could not be relied on [281,282]. The currently available anti-COVID drugs may hypothetically rather do harm than good; a debating subject needed to be clarified precisely over the course of time [283,284,285,286,287]. As a matter of fact it seems that COVID-19, beyond being a seemingly ineradicable infectious disease, is rather a dys-homeostatic phenomenon initiated exactly as the virus rushes to penetrate into the cell through exploiting the homeostatic regulatory pathways. Hindering the virus cell entry seems to be an astonishing task to do in abolishing the infectivity and simultaneous stopping emergence of newly mutated variants of the virus. In this context, rebalancing the signaling pathways derailed in the course of the virus entrance by repurposing safe drugs and products is worth noticing as it plays a pivotal role in surviving the patients without considering the presence or absence of the virus itself or whether it is mutated or not.

## 15. Conclusions

SARS-CoV-2 cell entry, a meticulously orchestrated process, necessitates that elaborate ongoing interactions be regulated among multiple factors. The virus, by inducing host cell proteases, facilitates its fusion with the plasma membrane and, vice versa, the effector proteins of the cell play a crucial regulatory role in promoting required conformational changes of the virus key molecules. Among those, the mutual interactions of the S protein and mACE2, both as nanomachines, are of paramount importance, blocking of which suppresses the virus infectivity. Moreover, the role of the cell key proteinases, TMPRSS2, furin, Cathepsins as well as ADAM17, in different models of SARS-CoV-2 cell entry should not be ignored. Inhibiting these interacting molecules may limit the pathological reactions of COVID-19. Additionally, it is also noticeable that the disordered homeostatic signaling cascades such as RAS, GPCR-mediated and protease-dependent pathways may aggravate the clinical course of SARS-CoV-2 infection, rebalancing of which may restrict the pathogenesis and hence the mortality as well as morbidity of the disease. There are a growing number of studies showing the abnormal predominance of pro-inflammatory ACE/Ang II/AT1R/Nox over anti-inflammatory ACE2/Angiotensin(1–7)/MasR pathways as the probable cause of chaotic inflammatory responses in COVID-19. In this context, regulating effects of ACE inhibitors and ARBs have undergone variable in vitro and in vivo studies followed by clinical trials with controversial results. Similarly, administration of rhsACE2 which, at first glance, may act as a vehicle enabling SARS-CoV-2 to spread to other tissues enhancing the severity of the disease, may be promising in blocking the complexification of S protein with mACE2 while could also return the consequences of the pathological disorganized pathways into normal organized cascades to recover tissues physiological functions.

## Figures and Tables

**Figure 1 vaccines-11-00204-f001:**
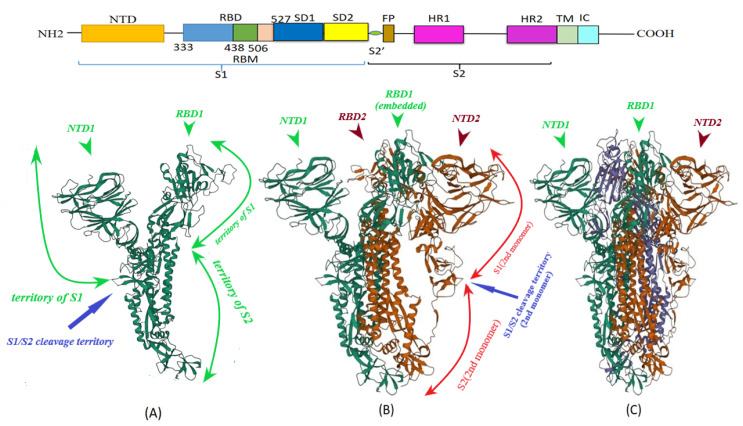
SARS-CoV-2 spike protein homotrimeric structure (**C**) (PDB ID: 7QUS [28]). Monomers (visualized as green, red and blue structures) are added one by one, from (**A**) to (**C**), to show how each monomer is folded and the dimeric as well as the trimeric structures are inter-twisted. It makes boundary identification of S1/S2 interface in a three-dimensional view difficult to show, yet has been depicted here by green curve arrows for simplification. The territory of S1/S2 cleavage site is also pointed (blue arrow). It is noticeable how in “closed, D” conformation (see the text, Section 3), RBD in the first monomer (**A**) is embedded in other S protein residues when the second (**B**) and third (**C**) monomers are added. This complex feature of S protein molecule leads RBDs to pack together as inaccessible domains in the “D” conformation (see the text below). NTD: N-terminal domain; RBD: receptor binding domain.

**Figure 2 vaccines-11-00204-f002:**
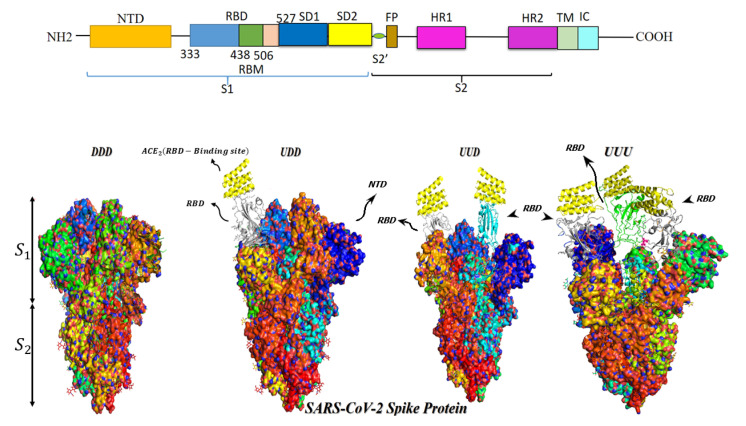
Linear representation of different domains of monomeric S protein and “open, U” and “closed, D” conformations of S protein. The defining of the boundary of S1 and S2 subdomains (arrows on the left side) is just a simplification (see Figure 1). Sequential opening of RBDs of the trimeric S protein after docking of the first open RBD to ACE2 is displayed. It results in an untwisting movement of the monomers away from each other after binding of each “U” RBD to ACE2 (see the text). Green helices at the top of S protein represents RBD-binding site of ACE2. NTD: N-terminal domain; RBD: receptor binding domain; RBM: receptor binding motif; SD1 and SD2: subdomains 1 and 2; FP: fusion peptide; HR1 and HR2: heptad repeat 1 and 2; TM: transmembrane domain; IC: cytoplasmic domain; DDD: three closed RBDs (PDB ID: 7UB5 [39], PyMol molecular graphics system); UDD: one open/two closed RBDs (PDB ID: 7KEB, [40], PyMol molecular graphics system); UUD: two open/one closed RBDs (PDB ID: 6X2B [41], PyMol graphics system); UUU: three open RBDs (PDB ID: 8CSA [42], PyMol molecular graphics system).

**Figure 3 vaccines-11-00204-f003:**
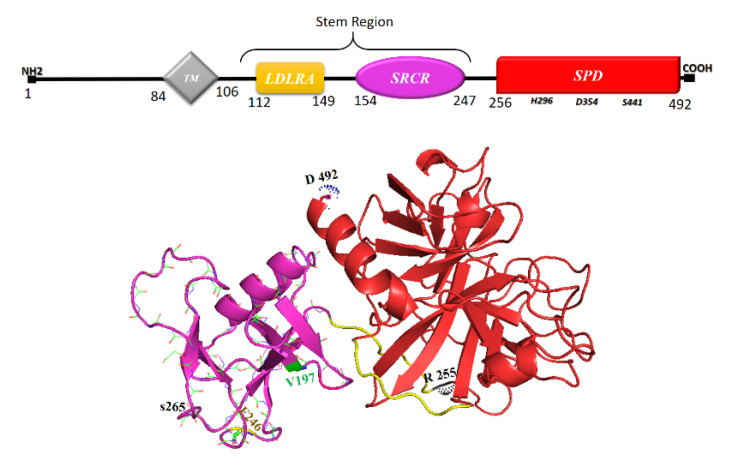
Linear representation of distinct domains and three dimensional shape of human TMPRSS2 (isoform 2) (PDB ID: 7MEQ [18], PyMol molecular graphics system).

**Figure 4 vaccines-11-00204-f004:**
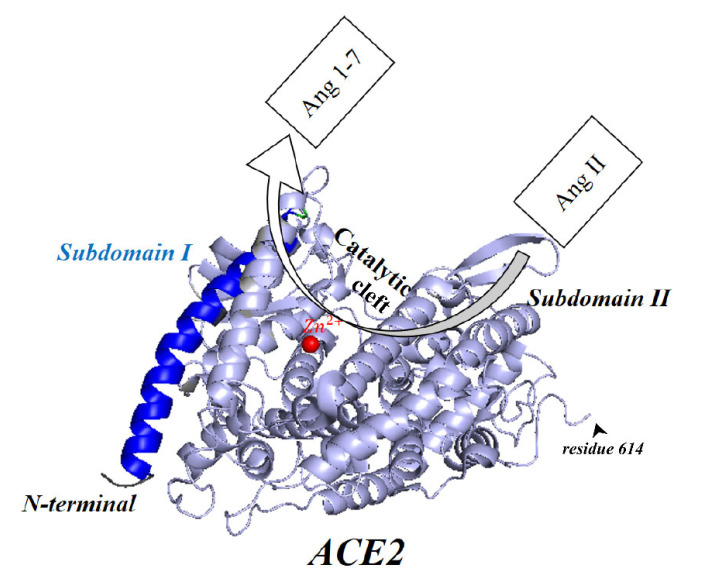
Three dimensional structure of the protease domain of ACE2 (PDB ID: 6M0J [24], PyMol molecular graphics system). The catalytic cleft, between subdomains I and II, involved in converting Ang II to angiotensin (1–7), is noticeable. The N-terminal helix in blue is the territory of binding of ACE2 with S protein. Residue 614 is the point of attachment of the catalytic domain on the way to C-terminal with other domains of ACE2; the neck domain, the transmembrane domain and the cytoplasmic tail.

**Figure 5 vaccines-11-00204-f005:**
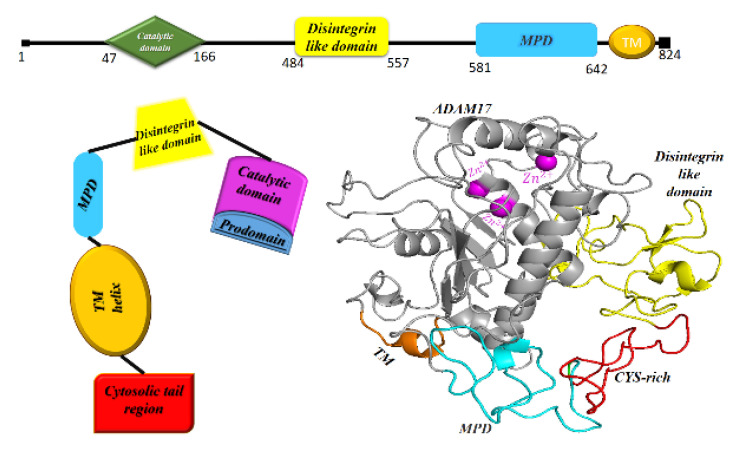
Linear representation of distinct domains, schematic and three dimensional shape of ADAM17 (PDB ID: 2DDF [164], PyMol molecular graphics system). Cys-rich: cysteine rich domain; MPD: membrane proximal domain; TMD: transmembrane domain.

**Figure 6 vaccines-11-00204-f006:**
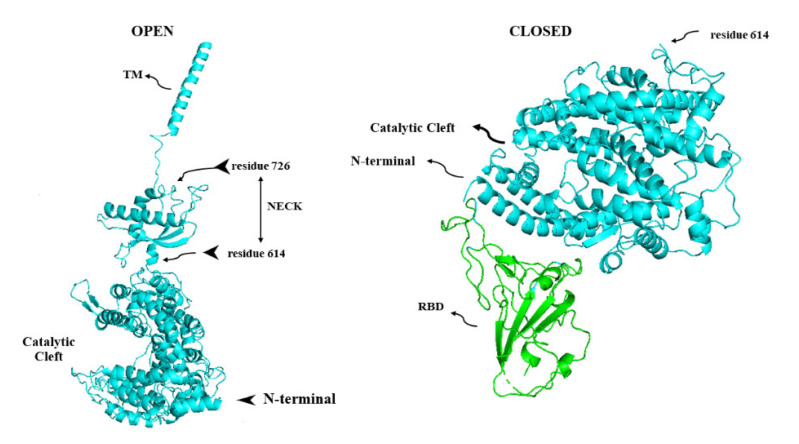
Three dimensional structure of open (catalytically active full length, greenish-blue) (PDB ID: 6M1D, [247], PyMol molecular graphics system) and closed (catalytically inactive, greenish-blue) (PDB ID: 7RPV, [248], PyMol molecular graphics system) forms of ACE2. As has been debated, complexification of RBD (light green) and ACE2 protease domain (the “closed” figure) makes the catalytic cleft shrink to some degree. RBD: receptor binding domain; TM: transmembrane domain.

**Figure 7 vaccines-11-00204-f007:**
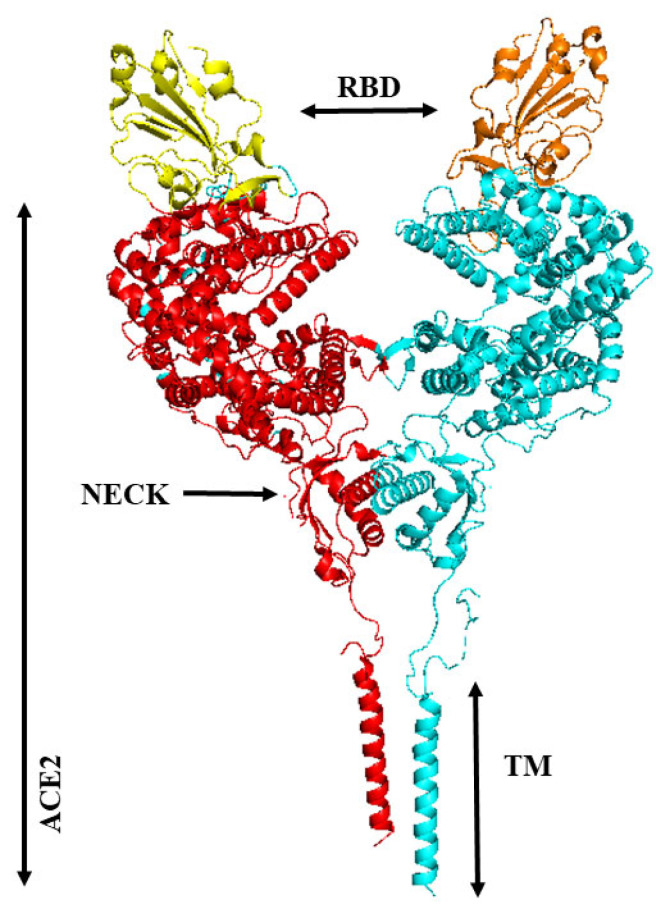
The three dimensional structure of an ACE2 homodimer molecule bound with two RBDs (yellow and orange) (PDB ID: 6M17, [247], PyMol molecular graphics system). ACE2 monomers are in red and greenish-blue colors. The “Neck” region is involved in homodierization of monomeric form of ACE2 molecules. PD: ACE2 peptidase domain, RBD: receptor biding domain of SARS-CoV-2 S protein, TMD: ACE2 trans-membrane domain.

## Data Availability

No new data was created in this study.

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
