# Peer review of "S Protein, ACE2 and Host Cell Proteases in SARS-CoV-2 Cell Entry and Infectivity; Is Soluble ACE2 a Two Blade Sword? A Narrative Review"

_vaccines, 2023, doi:10.3390/vaccines11020204_

Round 1
Reviewer 1 Report
The first and crucial step for SARS-CoV2 infectivity and pathogenesis is the entry into the host cells. This involves the physical interaction between Spike (S) protein of viruses and ACE2 membrane protein on surface of host cells. This interaction is regulated and controlled by multiple proteases of host cells, including Furin, TMPRSS2, Cathepsins, metalloproteases. In this review, Nejat and colleagues discussed the molecular mechanisms underlying the complex process of SARS-CoV2 entrance into the host cells and how this process is modulated by other proteins, in particular proteases of host cells. The authors also discussed soluble form of ACE2, generated by catalytic ectodomain cleavage, and its positive and negative impact on the virus infectivity. Overall, this is a very informative and well-organized review. I read it with lots of joys and strongly recommend for publication. I have a few minor points for the authors to improve the readability of the manuscript.
1. Page 2, paragraph 2: It should be "Ca2+-dependent" instead of "Ca+2-dependent". The authors should go through the whole manuscript for the same issue.
2. It would be nice to include a Figure to show the structure of Spike protein with domains annotated. I also suggest the author to present a published structure of the complex of ACE2 and S1 proteins. As a general rule, the authors may want to use figures to help readers understand the massive information in the main text.
3. The last sentence in the second paragraph of page 5 is ambiguous. Please rephrase for clarity.
4. The last sentence in the third paragraph of page 5 is too long and is difficult to understand. It would be better to split into two or more short sentences.
5. Page 7, paragraph 2, line 5: add a space into "rolein".
6. page 11, paragraph 4, last sentence: it should be "In order to block..." instead of "in order to blocking...".
Author Response
Dear Reviewer (Madam/Sir),
Happy New Year, 2023! Thank you for your reviewing the manuscript (vaccines-2102914) and your kind supporting report. I wonder if the figures which have been added would suffice; another figure showing S1 attached to ACE2 with annotations is going to be visualized. I would also appreciate if you could determine which sentences should be rewritten for more clarification of the subjects on page 5: please give me the clue which sentences.
Best wishes for you in 2023,
Reza Nejat, M. D.,
FCCM

Reviewer 2 Report
Estimated Authors,
thank you for the opportunity to read and review this very well organized, written and iconographically well sustained review on some features of SARS-CoV-2 infection.
From my point of view, Authors have performed a wonderful piece of work in summarizing a lot of of often contradictory information on molecular mechanisms we don't fully understand, and therefore this paper is clearly worthy to be published in Vaccines ASAP.
Still, two minor issues that could be fixed in post-acceptance stage, during the editorial managing of the paper:
1) please double check all the acronyms (for instance, you consistently write "SARS-CoV2" of SARS-CoV-2; please check the right one)
2) please check some minor misspeling (e.g. furin is sometime improperly written, but it clearly a minor typo)
3) please include in the main title the definition of the present paper as a narrative review (e.g. "S protein, ACE2 and host cell proteases in SARS-CoV2 cell entry and infectivity; Is soluble ACE2 a two blade sword? A narrative review")
4) please change a litttle bit the very first sentence of the paper "Deciphering the secret pattern of SARS-CoV2 " --> "Deciphering the pattern of SARS-CoV2"
Thank you for this very interesting paper.
Author Response
Dear Reviewer (Madam/Sir),
Happy New Year, 2023! Thank you for your reviewing the manuscript (vaccines-2102914) and indeed for your kind supportive report. Please have a look at the attached file.
Best wishes for you in 2023,
Reza Nejat, M. D., FCCM
